# Turing's children: Representation of sexual minorities in STEM

Dario Sansone[1]⊚, Christopher S. Carpenter[2]⊚*

**1** Department of Economics, Business School, University of Exeter and IZA, Exeter, United Kingdom,
**2** Department of Economics, Vanderbilt University, NBER and IZA, Nashville, Tennessee, United States of America

⊚ These authors contributed equally to this work.
* christopher.s.carpenter@vanderbilt.edu

**Data Availability Statement:** All data used in this work can be downloaded from IPUMS. The American Community Survey can be accessed here: https://usa.ipums.org/usa/. The National Health Interview Survey can be accessed here: https://nhis.ipums.org/nhis/.

## Abstract

We provide nationally representative estimates of sexual minority representation in STEM fields by studying 142,641 men and women in same-sex couples from the 2009–2018 American Community Surveys. These data indicate that men in same-sex couples are 12 percentage points less likely to have completed a bachelor's degree in a STEM field compared to men in different-sex couples. On the other hand, there is no gap observed for women in same-sex couples compared to women in different-sex couples. The STEM degree gap between men in same-sex and different-sex couples is larger than the STEM degree gap between all white and black men but is smaller than the gender gap in STEM degrees. We also document a smaller but statistically significant gap in STEM occupations between men in same-sex and different-sex couples, and we replicate this finding by comparing heterosexual and gay men using independently drawn data from the 2013–2018 National Health Interview Surveys. These differences persist after controlling for demographic characteristics, location, and fertility. Finally, we document that gay male representation in STEM fields (measured using either degrees or occupations) is systematically and positively associated with female representation in those same STEM fields.

## Introduction

In this paper, we provide the first nationally representative estimates of the representation of sexual minorities in Science, Technology, Engineering, and Mathematics (STEM) degrees and occupations. By doing so, we start to address the dire need for statistics on sexual and gender minorities in STEM emphasized in the letters sent to the National Science Foundation (NSF) by 251 scientists, engineers, legal and public policy scholars, as well as 17 scientific organizations [1,2].

Despite improvements in the legislative and institutional background for LGBTQ people, such as the legalization of same-sex marriage in numerous countries in the last twenty years, the workplace environment for LGBTQ scientists is still far from welcoming. Until a United States Supreme Court decision in 2020 (*Bostock v. Clayton County*), it was legal to discriminate

**Funding:** The authors received no specific funding for this work.

**Competing interests:** The authors have declared that no competing interests exist.

against applicants and employees based on their sexual orientation or gender identity in 25 states [3]. While the NSF tracks the participation rates of women, racial and ethnic minorities, and persons with disabilities in science and engineering [4], it does not routinely collect statistics on LGBTQ people. Other federal agencies, such as the National Institutes of Health, have historically funded only a very small fraction of LGBTQ-related projects [5]. Researchers have documented under-representation and worse workplace experiences for LGBT employees in STEM-related federal agencies [6]. In addition, several studies and reports have documented the academic and social isolation, as well as the heterosexist and uncomfortable workplace climate faced by LGBTQ STEM professionals [7–12], in addition to explicit anti-LGBTQ harassment [13–15]. Similar experiences have been documented in the medical profession [16,17].

Prior research has documented the presence of substantial gaps in STEM degree completion and occupational attainment in STEM fields associated with gender and race/ethnicity [4,18,19]. However, to our knowledge, there have been only a handful of studies (mostly based on non-random samples) on STEM representation for sexual minorities [8,10,20,21], in addition to general analyses of human capital accumulation by sexual orientation [22]. In particular, one prior study [23] used data from a 2015 survey of undergraduates containing 147 self-identified gay men: it found that, conditional on reporting a STEM major aspiration upon college entry, gay men were 14 percentage points less likely than straight men to persist in STEM majors by the fourth year of college (even if they were more likely to have worked in a lab).

Our study builds on this prior work in two critical ways. First, we use samples of sexual minorities that are two orders of magnitude larger than previous STEM studies. Specifically, we draw on data from the 2009–2018 American Community Surveys (ACS) which identify over 142,000 individuals in same-sex cohabiting romantic relationships. Moreover, the ACS contain information on the undergraduate major(s) for individuals who obtained bachelor's degrees, as well as detailed information on current occupation.

Second, we complement the ACS with evidence from the 2013–2018 National Health Interview Surveys (NHIS) which also contain detailed information on occupation as well as direct individual-level questions about sexual orientation. For example, this allows us to examine whether sexual minority representation in STEM fields differs between lesbian and bisexual women (including singles). Sample sizes in the NHIS are smaller than in the ACS, though they are still an order of magnitude larger than prior work (4,763 self-identified sexual minorities in the 2013–2018 NHIS).

## Materials and methods

### The American Community Surveys (ACS)

The main dataset used in our analysis is the ACS. The ACS is a nationally representative and repeated cross-sectional survey conducted by the U.S. Census Bureau. We use the ACS combined annual (1-year) estimates for each year from 2009 through 2018. These data contain demographic, economic, social, and housing information on 1 percent of the U.S. population (or approximately 3 million people each year). Such large sample sizes facilitate studies on relatively small subpopulations, such as individuals in same-sex couples and/or working in STEM occupations, or even heterogeneity analyses among these subgroups (e.g., by sex or race within same-sex couples). These data are publicly available through IPUMS-USA at the University of Minnesota [24].

The ACS does not directly ask individuals about their sexual orientation. To identify sexual minorities, we follow a large body of prior research that uses intrahousehold relationships to identify individuals in same-sex couples [25]. Specifically, the ACS identifies a primary reference person, defined as "the person living or staying here in whose name this house or

apartment is owned, being bought, or rented". For each individual in the household, the ACS also collects information on their sex and the individual's relationship to the primary reference person, and the range of possible relationships includes husband, wife, and unmarried partner (as a different category than roommate or other nonrelative). Thus, we identify individuals in same-sex couples in the following way: households with an adult who is the same sex as the primary reference person and whose relationship to the primary reference person is described as spouse or unmarried partner. A large body of research in social science and demography confirms that the vast majority of same-sex couples in the ACS are gay men and lesbians [26].

We restrict our attention to individuals age 18 to 65 who were interviewed between 2009 and 2018. We study the ACS data collected between 2009 and 2018 because information on the bachelor's degree field of study is available starting in 2009. Moreover, the U.S. Census Bureau implemented several changed between 2007 and 2008 to address concerns about misclassification errors and to increase data quality [27]. In addition, observations with imputed sex or relationship to the primary reference person have been dropped to further reduce measurement errors [28]. Our final sample includes 73,000 women and 69,641 men in same-sex couples, as well as 10,809,885 men and women in different-sex couples.

## The National Health Interview Surveys (NHIS)

The main disadvantage of using ACS data is that it is not possible to identify single LGBQ individuals without a partner or same-sex couples who do not live together. Furthermore, since there is no individual-level information on sexual orientation, researchers cannot identify sexual minority individuals in different-sex couples (e.g., a bisexual woman married with a man). In order to address these limitations, we have analyzed data from the NHIS. The NHIS is a household, face-to-face health survey conducted by the National Center for Health Statistics of approximately 87,500 people in 35,000 households each year. The NHIS sample is designed to be representative of the U.S. civilian, non-institutionalized population. Interviewers collect information from family reference adults on the household, socio-demographic characteristics, and health indicators for all persons in the selected households. In addition, extensive information (including employment status and occupation) is collected on one randomly selected sample adult and one sample child from each family. These data are publicly available through IPUMS-Health Surveys at the University of Minnesota [29].

From 2013, sample adults were asked whether they identified as straight, gay/lesbian, bisexual, or "something else". Between 2013 and 2018, our final sample with information on self-reported sexual orientation and occupation includes 67,367 heterosexual women (age 18 to 65), 59,732 heterosexual men, 1,213 lesbian\gay women, 1,524 gay men, 1,113 bisexual women, and 426 bisexual men. The sample also includes 279 women and 208 men who identified with another sexual orientation category.

## Terminology and STEM definitions

Throughout, we use the term "sexual minorities" to refer to individuals who describe themselves as lesbian, gay, bisexual, queer, or "something else". We also refer to this population as "LGBQ" for lesbian, gay, bisexual, and queer. Unfortunately, due to data limitations, we are unable to study transgender individuals (i.e., people whose gender identity and/or expression does not match their sex assigned at birth). Some studies in the literature [6] use data that include both sexual minorities and gender minorities; in those cases, we refer to the LGBTQ population (i.e., including transgender individuals).

We identify two key measures of representation in STEM fields in the ACS and NHIS: STEM degrees and STEM occupations. Information on STEM degrees is only available in the

ACS; respondents were asked to identify the specific major of any bachelor's degrees each individual in the household had received. Among individuals with a bachelor's degree, we code fields of study as being in STEM based on the individual's primary or first bachelor's degree. It is worth noting that the ACS measures degree completion, while [23] studied persistence in STEM by the fourth year of college but did not directly observe degree completion. STEM occupations are instead observed in both our datasets. We code occupations as being in STEM based on the individual's primary occupation.

As explained in detail in the Supporting Information, we follow the Department of Commerce and Bureau of Labor Statistics definitions to determine which degrees and occupations are in STEM fields. There are several reasonable alternative definitions of STEM degrees and STEM occupations. For example, some scholars include economics and finance degrees and professions in STEM. For our core definitions we do not code degrees in health, economics, or finance as STEM degrees. We also do not include teachers, health and medical professions, or economic and finance professions in the definition of STEM occupations.

The main STEM degree categories include: agricultural sciences; environmental science; architecture; communication technologies; computer and information systems; general engineering; engineering technologies; biology; mathematics; military technologies; interdisciplinary and multi-disciplinary studies (including nutrition science and neuroscience); physical sciences; nuclear, industrial radiology, and biological technologies; transportation sciences and technologies; actuarial science; operations, logistics and E-Commerce; and management information systems and statistics.

The main occupation categories from which STEM occupations are drawn include: STEM management occupations; computer and mathematical occupations; architecture and engineering occupations; life and physical science occupations; and sales engineers. Results with alternative definitions can be found in S1 Table.

## Methodology

We start by presenting descriptive statistics and mean comparisons in Tables 1 and 2 and visually in Figs 1 and 2. The ACS sample in Table 1 includes individuals (age 18–65) in a same-sex or different-sex couples. This table includes both the household primary reference person and the unmarried partner or married spouse in same-sex or different-sex couples. Some individuals age 18–65 may be partnered with individuals younger than 18 or older than 65, thus the sample size for men and women in different-sex couples is different. The NHIS sample in Table 2 includes all sample adults (age 18–65) who were working in the week preceding the interview, with a job or business but not at work, or who had ever worked. Respondents not in the universe, who refused to answer the occupation question or with missing information have been excluded. All reported statistics are weighted using survey sample weights.

We then report estimates from ordinary least squares models in Table 3. We report the coefficient on the sexual minority variables, and in each case the relevant excluded category is the dummy variable for the majority group (individuals in different-sex couples in the ACS, self-identified heterosexual individuals in the NHIS). In line with the statistics reported in Tables 1 and 2, the dependent variable in columns 1–2 of Table 3 is whether an individual received a bachelor's degree in a STEM field. The dependent variable in columns 3–6 of Table 3 is whether an individual used to work in a STEM occupation. To address retention and persistence in STEM, in Table 4 we report the results from a regression on the ACS data where the outcome is STEM occupation and where the sample is restricted only to individuals with a bachelor's degree in STEM.

**Table 1. Men (but not women) in same-sex couples are significantly less likely to have STEM degrees and work in STEM occupations than those in different-sex couples (ACS 2009–2018).**

|  | Women | | Gap | Men | | Gap |
|---|---|---|---|---|---|---|
|  | In same-sex couples | In different-sex couples |  | In same-sex couples | In different-sex couples |  |
| Bachelor's degree or more | 0.448 | 0.359 | 0.089*** | 0.487 | 0.339 | 0.148*** |
| STEM degree | 0.140 | 0.139 | 0.001 | 0.228 | 0.348 | -0.120*** |
| STEM occupation | 0.050 | 0.032 | 0.018*** | 0.084 | 0.095 | -0.011*** |
| Observations | 73,000 | 5,572,796 |  | 69,641 | 5,237,089 |  |

Weighed statistics using person weights. See also Data and Methodology. All variables are defined in detail in the SI. "Observations" refers to the total number of respondents in the relevant sub-group. Source: ACS 2009–2018.

* $p < 0.10$,

** $p < 0.05$,

*** $p < 0.01$

All regressions include controls for demographic characteristics (age, race, ethnicity), fertility (including the presence of any children in the household and any children under age 5 in the household), and location (state fixed effects in the ACS and region fixed effects in the NHIS since we do not observe state of residence in the NHIS public-use data). We estimate standard errors that are robust to heteroscedasticity, and we use the survey sample weights throughout. We account for the NHIS complex sample design by using the command *svy* in Stata 15 to include information on primary sampling units and strata.

It is important to emphasize that we are not accounting in our analysis for different selection into higher education and employment by sexual orientation or couple type. Indeed, as shown in Table 1 and S1 Table, individuals in same-sex couples have different levels of education, labor force participation, and employment than individuals in different-sex couples. Moreover, it is possible that certain sub-groups, e.g. low-income individuals or racial minorities, might be less likely to self-identify as members of a same-sex couple (they could for instance select the option "roommate" instead of "unmarried partner"). Therefore, we are not claiming that the results in Tables 3 and 4 have any causal meaning: we are only presenting estimates conditional on demographic characteristics, fertility and location, while we are not controlling for the fact that LGBQ individuals who get a bachelor's degree or enter into the labor force might be systematically different than heterosexual individuals.

**Table 2. Gay men are significantly less likely to be in STEM occupations than heterosexual men (NHIS 2013–2018).**

|  | Women | | | Men | | |
|---|---|---|---|---|---|---|
|  | N | STEM occupation | Gap with straight women | N | STEM occupation | Gap with straight men |
| Straight | 67,367 | 0.030 |  | 59,732 | 0.088 |  |
| Lesbian or gay | 1,213 | 0.034 | 0.004 | 1,524 | 0.065 | -0.023*** |
| Bisexual | 1,113 | 0.037 | 0.007 | 426 | 0.077 | -0.011 |
| Something else | 279 | 0.027 | -0.003 | 208 | 0.067 | -0.021 |

Weighed statistics using person weights and accounting for survey design. See also Data and Methodology. All variables are defined in detail in the SI. Source: NHIS 2013–2018.

* $p < 0.10$,

** $p < 0.05$,

*** $p < 0.01$

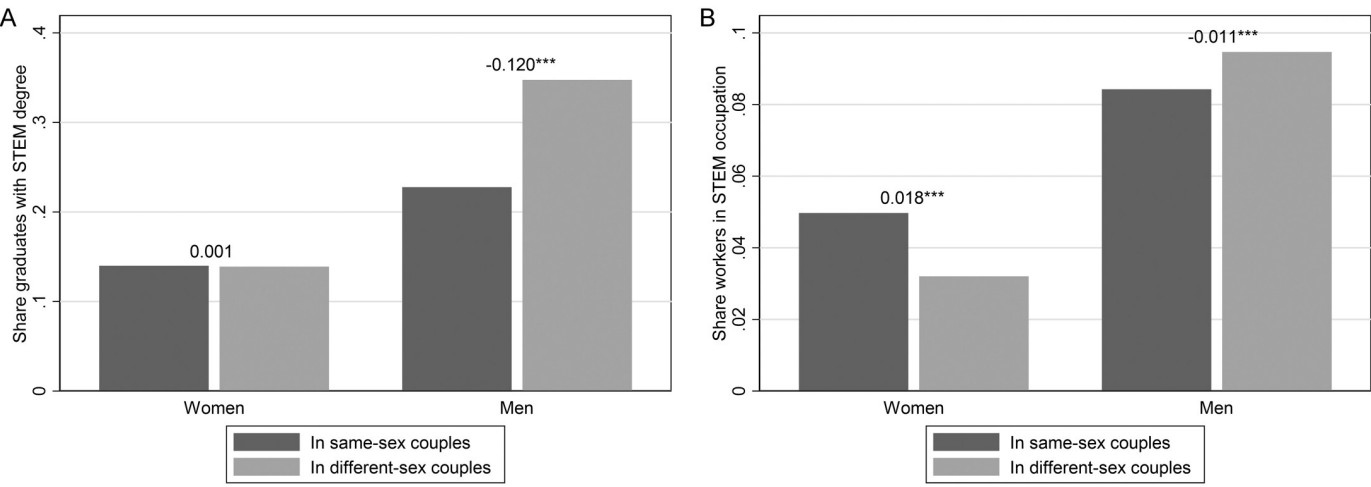

**Fig 1. STEM degree and STEM occupation gaps between individuals in same-sex couples and different-sex couples (ACS 2009–2018).** Panel A. STEM degree. Panel B. STEM occupation. The number above each bar is the gap between the share of male/female graduates/workers in same-sex couples vs. in different-sex couples in STEM degrees/occupations. Weighed statistics using person weights. See also Data and Methodology, as well as Table 1. All variables are defined in detail in the SI. Source: ACS 2009–2018. * $p < 0.10$, ** $p < 0.05$, *** $p < 0.01$.

We then analyze in Fig 3 the relationship between gay male representation in STEM fields (measured using either degrees or occupations) with female representation in those same STEM fields. Specifically, the x-axis in Panel A of Fig 3 is the share of individuals with bachelor's degrees in the STEM degree field that are women (of any marital status and relationship to the household primary reference person), and the y-axis is the share of coupled men with bachelor's degrees in the STEM degree that are men in same-sex couples (overall, 1.24% of

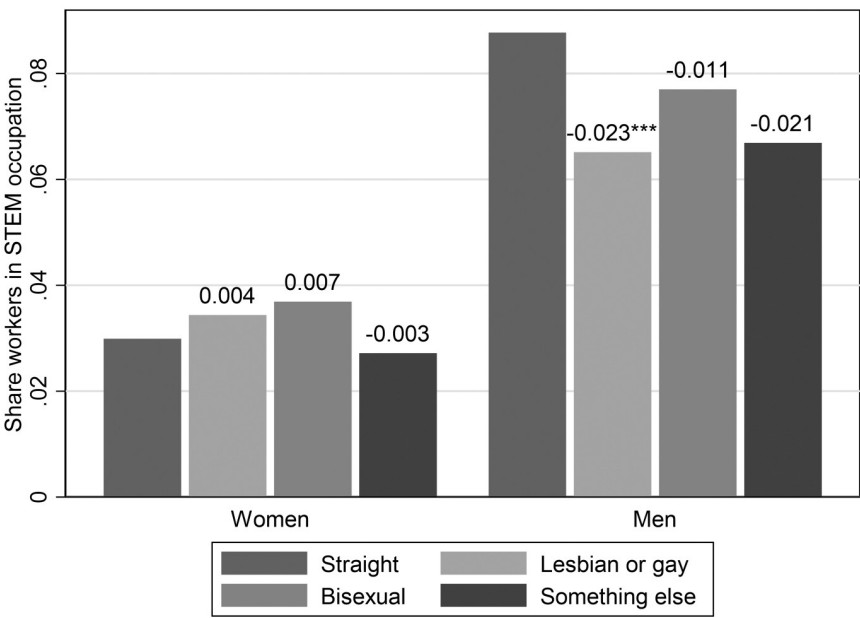

**Fig 2. STEM occupation gaps by sexual orientation (NHIS 2013–2018).** The number above each bar is the gap with respect to the share of straight male/female workers in STEM occupations. Weighed statistics using person weights and accounting for survey design. See also Data and Methodology, as well as Table 2. All variables are defined in detail in the SI. Source: NHIS 2013–2018. * $p < 0.10$, ** $p < 0.05$, *** $p < 0.01$.

**Table 3. STEM degree and STEM occupation gaps for gay men compared to heterosexual men are robust to controlling for demographic characteristics, fertility, and location.**

| | ACS 2009–2018 | | | | NHIS 2013–2018 | |
| --- | --- | --- | --- | --- | --- | --- |
| | STEM degree | | STEM occupation | | STEM occupation | |
| | Women | Men | Women | Men | Women | Men |
| | (1) | (2) | (3) | (4) | (5) | (6) |
| In a same-sex couple | 0.010*** | -0.106*** | 0.017*** | -0.016*** | | |
| | (0.002) | (0.003) | (0.001) | (0.001) | | |
| Gay or lesbian | | | | | 0.003 | -0.014* |
| | | | | | (0.006) | (0.008) |
| Bisexual | | | | | 0.006 | -0.014 |
| | | | | | (0.007) | (0.015) |
| Something else | | | | | -0.004 | -0.015 |
| | | | | | (0.014) | (0.020) |
| Dependent variable mean | 0.139 | 0.345 | 0.032 | 0.095 | 0.030 | 0.087 |
| R-squared | 0.030 | 0.039 | 0.015 | 0.029 | 0.013 | 0.029 |
| Observations | 2,063,090 | 1,850,340 | 4,664,190 | 4,992,047 | 69,972 | 61,890 |

The dependent variable in columns 1–2 is whether an individual received a bachelor's degree in a STEM field. The dependent variable in columns 3–6 is whether an individual used to work in a STEM occupation. See also Data and Methodology. All variables are defined in detail in the SI. All regressions include controls for demographic characteristics (age, race, ethnicity), fertility (indicators for children in the household and children under 5 in the household), and location (state fixed effects in the ACS, region fixed effects in the NHIS since we do not observe state of residence in the NHIS public-use data). Weighted regressions using person weights. Standard errors in parentheses. Source: ACS 2009–2018 and NHIS 2013–2018.

* $p < 0.10$,

** $p < 0.05$,

*** $p < 0.01$

**Table 4. STEM occupation gaps are larger when focusing on individuals with STEM degrees.**

| | ACS 2009–2018 (respondents with STEM degrees only) | |
| --- | --- | --- |
| | STEM occupation | |
| | Women | Men |
| | (1) | (2) |
| In a same-sex couple | 0.028*** | -0.093*** |
| | (0.008) | (0.006) |
| Dependent variable mean | 0.259 | 0.405 |
| R-squared | 0.043 | 0.037 |
| Observations | 252,827 | 622,282 |

The dependent variable is whether an individual used to work in a STEM occupation. Compare to columns 3–4 in Table 3. See also Data and Methodology. All variables are defined in detail in the SI. All regressions include controls for demographic characteristics (age, race, ethnicity), fertility (indicators for children in the household and children under 5 in the household), and state fixed effects. Weighted regressions using person weights. Standard errors in parentheses. Source: ACS 2009–2018 (respondents with STEM degrees only).

* $p < 0.10$,

** $p < 0.05$,

*** $p < 0.01$

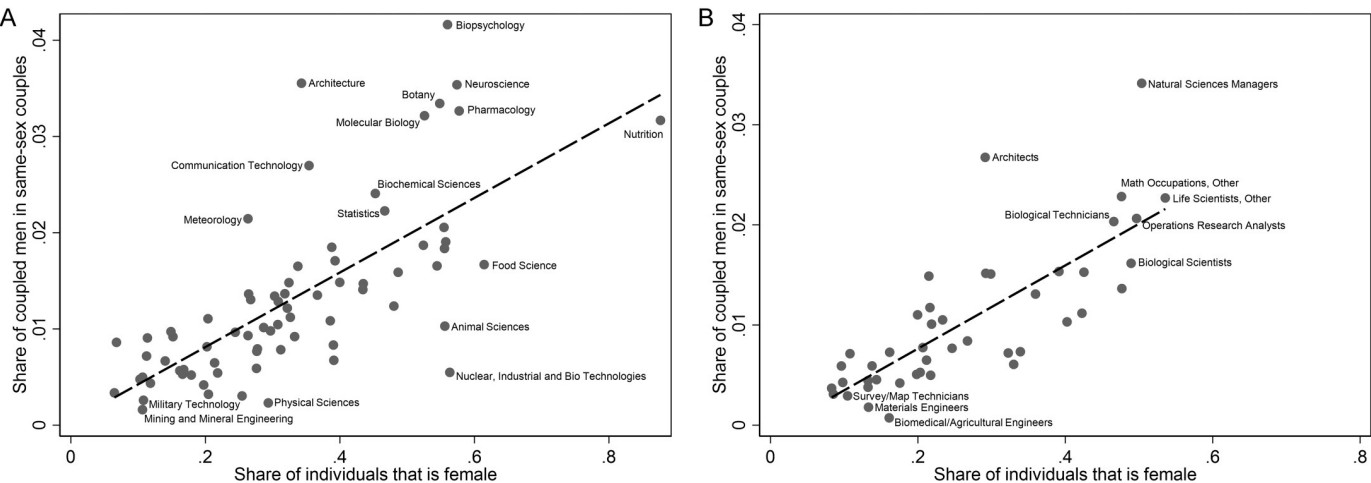

**Fig 3. There is a positive association between share of coupled men in same-sex couples and share women in STEM degrees and STEM occupations (ACS 2009–2018).** Panel A. STEM degrees. Panel B. STEM occupations. The vertical axis measures the share of men in same-sex couples over all coupled men in each field/occupation. Overall, 1.24% of men in a couple are in a same-sex couple. The horizontal axis measures the share of women (of any marital status and relation to the household head, age 18–65, sex not imputed) over all individuals in each field/occupation. Weighed shares using person weights. See also Data and Methodology. Only STEM fields/occupations reported. The dashed line plots the linear fit. Source: ACS 2009–2018.

men in a couple are in a same-sex couple). Each data point is a unique STEM degree field. We only report STEM fields. The dashed line plots the linear fit. Panel B of Fig 3 shows that the same relationship when focusing on STEM occupations rather than STEM degrees.

## Results

### Descriptive statistics

We begin by presenting the weighted means of our key variables separately by couple type in Table 1 (while S1 Fig shows how the gap in STEM fields and occupations between men in same-sex couples and men in different-sex couples varies geographically across the United States). Because of the large and well-documented gender gap in STEM, we present results separately for men and women. To provide context for the STEM degree gaps, we also report the share of each couple type with a bachelor's degree: women in same-sex couples are more likely to have a bachelor's degree than women in different-sex couples.

When focusing on STEM outcomes, it is evident that all women are underrepresented: women are always less likely to study or work in a STEM field, irrespective of their sexual orientation. In addition, there is essentially no gap among bachelor's degree holders in STEM degrees between women in same-sex couples and women in different-sex couples. When we examine STEM occupations, however, we observe that a larger share of women in same-sex couples are in STEM occupations than women in different-sex couples.

With respect to STEM degree attainment conditional on having a bachelor's degree, we find a notably different pattern for men from the one for women: there is a statistically significant gap in STEM degrees among men in same-sex couples and men in different-sex couples with bachelor's degrees. This gap is larger in size (12 percentage points) than the overall STEM gap between white and black men (4 percentage points) but is smaller than the gender STEM gap (21 percentage points). Because we can only identify sexual minorities in couples in the ACS data, we have compared the gap between individuals in same-sex couples and individuals in different-sex couples to other couples-based gaps (i.e., black men in couples versus white men in couples, and men in couples versus women in couples). The race and gender gaps in

STEM degrees are very similar if we consider all adults (i.e., if we do not restrict attention to individuals in couples), and the qualitative ordering remains true: the black/white gap in STEM degrees among men is smaller than the gap in STEM degrees between men in same-sex couples and men in different-sex couples, which itself is smaller than the gender gap in STEM degrees.

The lower rate of STEM degree attainment by men in same-sex couples with bachelor's degrees is particularly interesting in the context of their much higher rate of earning any type of bachelor's degree at all: despite being 43.6 percent *more* likely to have a bachelor's degree at all than men in different-sex couples, men in in same-sex couples with bachelor's degrees are 34.5 percent *less* likely to have completed that bachelor's degree in a STEM field than men in different-sex couples who earned a bachelor's degree.

The gap in STEM degrees between men in same-sex couples and men in different-sex couples is also observed for STEM occupations. Although the size of the STEM occupation gap by couple type for men is smaller, it is still statistically significant at the one percent level. We present these patterns visually in Fig 1.

S1 Table examines the sensitivity of the raw ACS STEM gaps presented in Table 1 to various alternative definitions of what constitutes a STEM degree or occupation. The patterns in S1 Table show that our patterns are largely unaffected by these choices, with the exception of health degrees and health professions. S2 and S3 Tables present the associated means for STEM degrees and STEM occupations additionally disaggregated by race and age groups, respectively. Asian people are much more likely to have STEM degrees and to work in STEM occupations than white or black individuals. Notably, the gap between individuals in same-sex and different-sex couples in STEM outcomes for women are slightly positive when looking at white or black women, while they are negative when focusing on STEM degrees among Asian women. The gap in STEM degrees and STEM occupations between Asian men in same-sex couples and Asian men in different-sex couples is much larger than the associated gaps between white/black men in same-sex couples and white/black men in different-sex couples. In addition, the gaps between individuals in same-sex and different-sex couples do not vary substantially across age groups.

We present the associated evidence on STEM occupations from the NHIS data in Table 2. None of the differences in STEM occupations between the self-identified non-heterosexual female groups and the heterosexual women is large or statistically significant.

For men in the NHIS, the gap between self-identified heterosexual and gay men in STEM is statistically significant and qualitatively identical to the ACS couples-based gap in STEM occupations documented in Table 1. We also observe that self-identified bisexual men and men who describe their sexual orientation as "something else" are less likely to be in STEM occupations than heterosexual men, though these differences in means are not statistically significant due to small sample sizes. Fig 2 presents the NHIS patterns visually.

## Multivariate analysis

In addition to documenting the size of the unadjusted gaps in STEM degrees and occupations by sexual orientation, it is also interesting to understand the extent to which these differences can be explained by differences across groups in observable characteristics. In Table 3, we examine whether the differences in STEM degrees and STEM occupations persist once we control for age, race, ethnicity, location, and fertility.

The resulting patterns largely confirm that the unadjusted gaps in STEM outcomes survive adjustment for the aforementioned observable characteristics. For example, for men in same-sex couples compared to men in different-sex couples, the raw gap documented in Table 1 of

12 percentage points falls slightly to 10.6 percentage points once we adjust for demographic characteristics, fertility, and location (column 2), though this estimate remains statistically significant at the one percent level. The patterns for STEM occupations in columns 3 and 4 are qualitatively similar: we continue to find that women in same-sex couples are slightly more represented in STEM occupations than women in different-sex couples, while the opposite is true for men in same-sex couples compared to men in different-sex couples.

Columns 5 and 6 of Table 3 perform the same regression adjustment exercise for self-identified sexual minorities in the NHIS. Here too we observe that the patterns observed in the raw differences in means are also observed in the regression estimates. Specifically, gay men are 1.4 percentage points less likely to be in STEM occupations than otherwise similar heterosexual men with the same age, race/ethnicity, fertility, and location, and this estimate is statistically significant at the ten percent level (column 6). None of the estimates on the other sexual minority indicators is statistically significant for women or for men due to the large standard errors, thus highlighting the relatively small sample sizes in the NHIS. Importantly, we note that both estimates comparing sexual minority men to heterosexual men across the ACS and NHIS are statistically significant, suggesting that there is a robust association between sexual orientation and STEM (under)representation for men in two independently drawn datasets. In line with the similar estimates for men in same-sex couples in the ACS and gay men in the NHIS, it is also worth mentioning that most men in same-sex couples identify as gay: bisexual men are more likely to be in different-sex couples [30].

While the differential magnitude between the STEM degree gap and the STEM occupation gap between men in same-sex couples and men in different-sex couples documented in Tables 1–3 may be at first surprising, it is explained by the much higher rates of bachelor's degree attainment by men in same-sex couples documented in the top row of Table 1. That is, while the difference in STEM degrees conditional on having a bachelor's degree between men in same-sex couples and men in different-sex couples is large (12 percentage points), the associated difference not conditional on having a bachelor's degree—i.e., counting all those without a college education as not having a STEM degree rather than excluding them from the analysis—is much smaller (0.7 percentage points) and thus similar in magnitude to the raw gap in STEM occupations between the two groups. We report the STEM degree gap conditional on having a bachelor's degree to be more consistent with existing literature and to emphasize that, even if more gay men might decide to enroll in college, they are still less likely to specialize in a STEM field. Similarly, Table 1 also highlights the presence of a large gender gap in STEM degrees and STEM occupations, and it indicates that far fewer people are in STEM occupations than are observed to have STEM degrees, a fact that has been previously documented [31].

In various analyses in the Supporting Information we probe the robustness and heterogeneity of the main findings. For example, different permutations in our set of controls result in qualitatively similar estimates (S4–S9 Tables). Our conclusions do not change also when including year fixed effects (S10 Table). The same broad pattern is true when we control for educational attainment in regressions predicting STEM occupations (S11 Table), though the magnitude of the gap between gay men and heterosexual men increases substantially. This is because, as shown in Table 1 and mentioned in the previous paragraph, gay men have significantly higher educational attainment than heterosexual men: controlling for these differences produces even larger estimated differences in STEM occupations (since education is positively related to the likelihood of working in STEM). We can further test the stability of the estimated gaps reported in Table 3 between men in same-sex couples and men in different sex couples by following [32]. Oster's method and suggested calibration implies that the unobservables would need to be 19 times as important as the observables to push the gap in STEM degrees between men in same-sex and men in different-sex couples (column 2 Table 3) to 0, well above the

heuristic threshold of 1. A similarly high value (10) is obtained when testing the robustness of the gap in STEM occupations between men in same-sex and men in different-sex couples (column 3 Table 3).

In Table 4 we investigate a related question related to the retention and persistence of STEM degree holders in STEM occupations. This question is interesting in part because women with STEM degrees are less likely to persist in STEM occupations [4], and one argument for this phenomenon is that unfriendly work environments contribute to the lack of persistence of women in STEM fields. Therefore, the same patterns could emerge when focusing on sexual minorities. We note that S1 Table does indicate that the raw gap in STEM degrees between men in same-sex couples and men in different-sex couples is much larger when we restrict the sample to individuals with STEM degrees than when we do not impose this sample restriction. Consistent with this, in Table 4 we show the results from our main regression-adjusted specification where we similarly restrict attention to STEM degree holders. Note that we can only do this in the ACS because we do not observe STEM degree status in the NHIS.

The patterns in Table 4 indicate that conditional on having a STEM degree, men in same-sex couples are significantly less likely to be working in a STEM occupation than otherwise similar men in different-sex couples, and this gap is much larger than the associated gap when we do not restrict the sample to STEM degree holders (9.3 percentage point gap in column 2 of Table 4 versus 1.6 percentage point gap in column 4 of Table 3). Overall, these raw and regression-adjusted patterns are consistent with the hypothesis that sexual minorities are disproportionately 'pushed out' of/not retained in STEM occupations even when they have the relevant STEM degrees.

Lastly, Fig 3 presents evidence that the mechanisms underlying the gender gap in STEM may be related to those driving the gap in STEM between gay and heterosexual men. Extensive research has documented a robust gender gap in STEM degrees and STEM occupations [4]. The data we analyze confirm that the gender gap in STEM is pervasive, affects both heterosexual and sexual minority women, and is larger than the associated gap between sexual minority men and heterosexual men. A natural question is whether the gap in STEM fields experienced by gay men is systematically related to the gender gap in STEM. Prior research has documented occupational sorting by gay men into female-dominated occupations [33]. Is this also the case in STEM?

There is a clear positive relationship in Panel A of Fig 3 between the share female in STEM degrees and the share of coupled men that is gay in STEM degrees. Moreover, Panel B of Fig 3 shows that the same relationship is observed for STEM occupations. S2 Fig shows that these positive relationships are unique to men in same-sex couples: there is no relationship (or a weakly negative one) observed when we plot the share of coupled women in STEM degrees or STEM occupations that are women in same-sex couples against the share of individuals in STEM that are women. S3 Fig shows the same pattern for men where we replace the individual data points with circles representing the size of the sub-samples of degree or occupation holders underlying each field and we weight the linear fit with these sample sizes. We present the data underlying Fig 3 and S3 Fig in tabular form in S12 and S13 Tables. Furthermore, S4 and S5 Figs perform the same exercise where we replace in the x-axis the share of women in the field with the share of black or African American men, and share of individuals with any disability (using the broadest ACS definition that includes ambulatory difficulty, independent living difficulty, cognitive difficulty, difficulty taking care of own personal needs, and vision or hearing difficulty), respectively. These figures show that the positive relationship documented in Fig 3 is unique to the share of women in the field. This finding for sexual minority men using large nationally representative ACS data confirms patterns in prior research using online

(nonrandom) samples that STEM fields with more representation of women are associated with an increased likelihood that LGBTQ people are open and out in those fields [20].

## Discussion

Taken together, these patterns are highly suggestive that the mechanisms underlying the very large gender gap in STEM fields such as heteropatriarchy [34], implicit and explicit bias, sexual harassment, unequal access to funding, and fewer speaking invitations [35] are related to the factors driving the associated gap in STEM fields between gay men and heterosexual men. For example, perceptions that gay men are relatively feminine and that lesbian women are relatively masculine may contribute in part to the underrepresentation of gay men compared to heterosexual men in STEM and the lack of differential representation of lesbians compared to heterosexual women in STEM. The patterns also suggest that policies to improve representation of women in STEM fields (e.g., reducing toxic masculinity) may have the associated benefit of increasing representation of gay men in STEM fields, and vice versa [20].

We hope that our findings will emphasize the importance of focusing on sexual orientation in addition to sex, race, ethnicity and disability when discussing the status of minorities in STEM fields. As with prior evidence on STEM gaps associated with gender and race, our findings on LGBQ-related STEM gaps are important not only for equity considerations, but also because addressing these gaps could increase efficiency by improving group decision-making, company performance, and the quality and variety of scientific work [36]. In addition, increasing the number of LGBQ people in STEM could help to alleviate the chronic shortage of workers in these fields [37]. Future research should also investigate in more detail the representation of sexual minorities in health-related degrees and occupations, as our results suggest that the underlying mechanisms and dynamics may be different in those fields.

While we cannot directly comment on STEM representation differences associated with gender identity due to data limitations, our work highlights the need for more large nationally representative data on both sexual and gender minorities in STEM to better understand their representation in undergraduate and graduate programs, in academia, and in the private sector, as well as the specific barriers and challenges faced by these groups. An important step—currently under discussion at the NSF [38]—would be to regularly include sexual orientation and gender identity measures in NSF surveys such as the Survey of Earned Doctorates, the Survey of Doctorate Recipients, and the National Survey of College Graduates [1,2].

Finally, there are several areas and best practices that have been identified to foster representation of LGBTQ members in STEM fields. Researchers have already emphasized the importance of role models, representation, community, and equal treatment from employers [11,39,40]. Campaigns such as *500 Queer Scientists* and associations such as the *National Organization of Gay and Lesbian Scientists and Technical Professionals* are actively increasing visibility and supporting LGBTQ STEM workers. Federal agencies and universities could include LGBTQ representation into their diversity objectives [36]. More generally, fostering the use of gender-neutral pronouns could lead to more positive attitudes towards women and LGBTQ individuals [41].

## Supporting information

**S1 Text. ACS variables description.**
(DOCX)

**S2 Text. NHIS variables description.**
(DOCX)

**S1 Fig. STEM degree and STEM occupation gaps between men in same-sex couples and men in different-sex couples, by state (ACS 2009–2018).** Panel A: STEM degrees. Panel B: STEM occupations. Notes: This map shows the gaps in STEM degrees and STEM occupations between men in same-sex couples and men in different-sex couples in each state. Darker colors indicate that the gaps between the share of men in same-sex couples and the share of men in different-sex couples working or with a STEM degree is smaller (or even positive). Weighed shares using person weights. See also Data and Methodology. Source: ACS 2009–2018.
(DOCX)

**S2 Fig. Relationship between share of coupled women in same-sex couples and share women in STEM degrees and STEM occupations (ACS 2009–2018).** Panel A: STEM degrees. Panel B: STEM occupations. Notes: The vertical axis measures the share of women in same-sex couples over all coupled women in same-sex or different-sex couples in each field/occupation. Overall, 1.27% of women in a couple are in a same-sex couple. The horizontal axis measures the share of women (of any marital status and relation to the household head, age 18–65, sex not imputed) over all individuals in each field/occupation. Weighed shares using person weights. See also Data and Methodology. Only STEM fields/occupations reported. The dashed line plots the linear fit. Source: ACS 2009–2018.
(DOCX)

**S3 Fig. Relationship between share of coupled men in same-sex couples and share women in STEM degrees and STEM occupations (ACS 2009–2018).** Weight by field/occupation size. Panel A: STEM degrees. Panel B: STEM occupations. Notes: Compare to Fig 3. See also Data and Methodology. The vertical axis measures the share of men in same-sex couples over all coupled women in same-sex or different-sex couples in each field/occupation. Overall, 1.24% of men in a couple are in a same-sex couple. The horizontal axis measures the share of women (of any marital status and relation to the household head, age 18–65, sex not imputed) over all individuals in each field/occupation. Weighted shares using person weights. Only STEM fields/occupations reported. Each circle is proportional to the number of degree holders or workers (of any marital status and relation to the household head, age 18–65, men and women, sex not imputed) in that specific field/occupation. The dashed line plots the linear fit using field/occupation sizes as weights. Source: ACS 2009–2018.
(DOCX)

**S4 Fig. Relationship between share of coupled men in same-sex couples and share black or African American men in STEM degrees and STEM occupations (ACS 2009–2018).** Panel A: STEM degrees. Panel B: STEM occupations. Notes: The vertical axis measures the share of men in same-sex couples over all coupled men in each field/occupation. Overall, 1.24% of men in a couple are in a same-sex couple. The horizontal axis measures the share of black or African American men (of any marital status and relation to the household head, age 18–65, sex not imputed) over all men in each field/occupation. Overall, 12.34% of men (of any marital status and relation to the household head, age 18–65, sex not imputed) are black or African American. Weighed shares using person weights. See also Data and Methodology. Only STEM fields/occupations reported. The dashed line plots the linear fit. Source: ACS 2009–2018.
(DOCX)

**S5 Fig. Relationship between share of coupled men in same-sex couples and share of individuals with any disability in STEM degrees and STEM occupations (ACS 2009–2018).** Panel A: STEM degrees. Panel B: STEM occupations. Notes: The vertical axis measures the share of men in same-sex couples over all coupled men in each field/occupation. Overall,

1.24% of men in a couple are in a same-sex couple. The horizontal axis measures the share of individuals (of any marital status and relation to the household head, age 18–65, men and women, sex not imputed) with any disability over all men in each field/occupation. Overall, 10.7% of individuals (of any marital status and relation to the household head, age 18–65, men and women, sex not imputed) have a disability coded in the IPUMS ACS as ambulatory difficulty, independent living difficulty, cognitive difficulty, self-care difficulty, vision or hearing difficulty. Weighed shares using person weights. See also Data and Methodology. Only STEM fields/occupations reported. The dashed line plots the linear fit. Source: ACS 2009–2018.
(DOCX)

**S1 Table. STEM degrees and occupations by type of couple.** ACS extensions.
(DOCX)

**S2 Table. STEM degrees and occupations by type of couple and by race.**
(DOCX)

**S3 Table. STEM degrees and occupations by type of couple and by age groups.**
(DOCX)

**S4 Table. STEM degree and STEM occupation gaps as in Table 3.** No controls.
(DOCX)

**S5 Table. STEM degree and STEM occupation gaps as in Table 3.** Control for age.
(DOCX)

**S6 Table. STEM degree and STEM occupation gaps as in Table 3.** Control for race/ethnicity.
(DOCX)

**S7 Table. STEM degree and STEM occupation gaps as in Table 3.** Control for demographics.
(DOCX)

**S8 Table. STEM degree and STEM occupation gaps as in Table 3.** No fertility controls.
(DOCX)

**S9 Table. STEM degree and STEM occupation gaps as in Table 3.** No location controls.
(DOCX)

**S10 Table. STEM degree and STEM occupation gaps as in Table 3.** Add year fixed-effects.
(DOCX)

**S11 Table. STEM degree and STEM occupation gaps as in Table 3.** Add higher education.
(DOCX)

**S12 Table. Fig 1 (STEM degrees) in tabular form.**
(DOCX)

**S13 Table. Fig 1 (STEM occupations) in tabular form.**
(DOCX)

## Acknowledgments

We are grateful to Cevat Aksoy, Jon Freeman, Luis Leyva, Andrew Penner, and Keivan Stassun for helpful comments.

## Author Contributions

**Conceptualization:** Dario Sansone, Christopher S. Carpenter.

**Data curation:** Dario Sansone.

**Formal analysis:** Dario Sansone.

**Investigation:** Dario Sansone, Christopher S. Carpenter.

**Methodology:** Dario Sansone, Christopher S. Carpenter.

**Writing – original draft:** Dario Sansone, Christopher S. Carpenter.

**Writing – review & editing:** Dario Sansone, Christopher S. Carpenter.

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
