## [Decision Letter · Decision Letter 0]

28 Sep 2020

PONE-D-20-24017

Turing’s Children: Representation of Sexual Minorities in STEM

PLOS ONE

Dear Dr. Carpenter,

Thank you for submitting your manuscript to PLOS ONE. After careful consideration, I feel that it has merit but does not fully meet PLOS ONE’s publication criteria as it currently stands. Therefore, I invite you to submit a revised version of the manuscript that addresses the points raised during the review process.

Reviewer #1 has made a number of suggestions that I believe would increase the value and impact of your analysis.  I am not able to assess the amount of time and effort that incorporating these suggestions would require, but ask you to consider whether you believe the benefits justify the additional costs.  Based on the evaluation of the reviewers I do not believe that a revised version of this article would require further peer review, but I want to be sure that you have given due consideration to Reviewer #1's suggestions.

We look forward to receiving your revised manuscript.

Kind regards,

Joshua L Rosenbloom

Academic Editor

PLOS ONE

Journal Requirements:

2. Please include captions for your Supporting Information files at the end of your manuscript, and update any in-text citations to match accordingly. Please see our Supporting Information guidelines for more information: http://journals.plos.org/plosone/s/supporting-information

Reviewers' comments:

Reviewer's Responses to Questions

**Comments to the Author**

1. Is the manuscript technically sound, and do the data support the conclusions?

Reviewer #1: Partly

Reviewer #2: Yes

Reviewer #3: Yes

2. Has the statistical analysis been performed appropriately and rigorously? 

Reviewer #1: Yes

Reviewer #2: Yes

Reviewer #3: Yes

3. Have the authors made all data underlying the findings in their manuscript fully available?

Reviewer #1: Yes

Reviewer #2: Yes

Reviewer #3: Yes

4. Is the manuscript presented in an intelligible fashion and written in standard English?

Reviewer #1: Yes

Reviewer #2: Yes

Reviewer #3: Yes

5. Review Comments to the Author

Reviewer #1: This study provides nationally-representative descriptive evidence on rates of STEM degree attainment and scientific workforce participation for sexual minorities. The paper is clear and well-written, and provides convincing evidence of occupational segregation by sexual orientation, usefully comparing the discrepancies found to previously-noted gaps in STEM participation by gender and race/ethnicity. However, given the importance of the topic, I offer some suggestions to strengthen the impact and presentation of their work, particularly to distinguish it from and build on each authors' previously published work on LGBT students in higher education.

1) ACS data

To better enable replication of the paper's approach, it would be helpful to have more detail on the version of the ACS data actually used, for example whether it combined annual (1-year) estimates for each year 2009 through 2018, or took advantage of 3- or 5-year estimates which allow incorporation of greater geographic detail. Since the models seem not to include year fixed effects, I'm presuming pooled estimates, but it would be helpful to know for sure.

I would also like to see more explicit discussion of the authors' decision to include state fixed effects in the ACS models, as it seems plausible there may be correlation between share of occupations in STEM fields and sociopolitical attractiveness of a region, due to e.g. presence of universities. If more or fewer STEM jobs are generally available in a region that is relatively unattractive for "out" same-sex couples, it seems like that could influence the results, to the extent that labor is mobile across regions. Sensitivity tests excluding state fixed effects, and excluding California in particular from the sample, would be a start.

2) What about the children?

One of the arguments for lesbian pay premium (which could be due to working in STEM occupations, which typically require greater human capital investment in higher education) is based on the notion that lesbians are less likely to "opt out" due to childcare. I suggest including a control in the regressions in Table 3 for presence of a child under age 6 as well as any own-child present in the home (as in appendix Table S3), as the apparent lack of impact in the appendix result was, to me, unexpected.

3) Retention of STEM grads in STEM occupations

For the subset of respondents who do earn STEM degrees, are LGBTQ respondents more or less likely to remain in STEM occupations? Women with bachelor's degrees in STEM fields are less likely than men to persist in STEM occupations, overall, though some of that may be partly driven by an artifact of categorization: health/medical occupations aren't considered STEM, whereas biology majors (popular among premeds) *are* considered STEM. One argument given for women's lower retention is unfriendly work environments as noted in the paper's Discussion section. It would be interesting to know whether LGBTQ graduates seem to be similarly pushed out (again controlling for childrearing). I'd also like to see some sensitivity checks around the definition of STEM occupations, specifically testing health sciences fields (think e.g. radiology technicians). I also think the inclusion of Nutrition (see Fig. 1) is a mistake; Nutrition degrees are sometimes in Education schools (think K-12 Health teachers) and don't necessarily require much lab science to complete.

4) Table 3 analysis

I understand the point of including the NHIS data was precisely to deal with the identification limitations of the ACS data; nonetheless, why not include results for the binary indicator "in a same-sex couple" for NHIS data with definition as close as possible to that in the ACS data, to see if the results are consistent?

In addition, it's not clear to me why the regressions presented in Table 3 are split by gender, particularly given inclusion of state and region fixed effects. If geography fixed effects are meant to account for e.g. demand-side differences in STEM fields or sociopolitical differences across states/regions, it seems reasonable to assume those effects would be similar for LGBTQ men and women. I would suggest running a single model for each outcome X data set, using indicators for gender and sexual orientation and their interactions to identify these differential effects while holding geographic effects constant. Including race/ethnicity X gender interactions would allow you still to control for intersectionality (as the separate regressions by gender implicitly allow) on those dimensions.

Also in Table 3, related to comment #3 above, it seems like it would be important to control for educational (degree) attainment, given that there are likely differences in degree attainment by sexual orientation, overall. For example, is the difference in STEM bachelor's degrees due to differences in bachelor's degree attainment overall? And if STEM occupations are more likely to require bachelor's degrees in STEM, how much of the difference in STEM occupations is driven by differences in bachelor's degree attainment, overall?

5) More visuals?

One powerful aspect of this journal is the opportunity to provide (more) color Figures to tell the story visually. I like the bottom panel of Figure 1, though I wonder if an exponential might be more appropriate than the linear fit suggested. I'm not sure if you'd have large enough N to exploit the geographic information this way, but something like a heatmap showing which states have larger discrepancies in LGBTQ STEM workforce participation could be interesting. I'd encourage the authors at minimum to consider adding graphical depictions of the descriptive results presented in tables 1 & 2.

Reviewer #2: This paper provides estimates of the gender, LGB, and racial make up of STEM Bachelors Degree recipients. The noted patterns are backed up While much has been said about the gender and racial composition of STEM fields, data on LGB content of them is sorely lacking. This general lack of information leaves those interested in the diversity of STEM fields without guidance – in order for policy makers at places like the NSF or NIH to begin to address sexual-orientation they need some basic description of the situation.

This paper is the sort of descriptive analysis that often I say I wish to see more of. It doesn’t follow the pattern that a student educated on well-placing JMP and top-5 articles has come to expect define an economics article; but it establishes basic facts, along with situating context, that one should not expect the author of a law review article, congressional report, or policy brief to calculate and establish for themselves. Thus, this paper contributes to the project of knowledge production for a wider audience.

I have one request that I feel strongly about and several thoughts on how to add a bit more to the exploration that is currently in the paper.

The graphs showing the relationship between the fraction of degree holders that are women and the fraction of coupled men in a same-sex couple are fascinating. However, I was left wondering how large each field was. Certainly there is a large amount of variation. I would like to see these graphs made with bubbles of varying size to represent the number of degree holders. Also please note if the linear fit takes into account this weighting. In addition, since this very interesting, releasing the information in tabular format so one can see which field are represented would be pleasing. I feel strongly that these things would improve the interpretability of this paper.

Related to the above, I was left wondering if there was a relationship between the fraction of gay men in a field and other measures of diversity. One could imagine that acceptance of feminine people in particular might mean that this field feels more welcoming to gay men, however, that this relationship does not translate to other minoritized groups. Is there a relationship between the fraction of Black men and men in same-sex couples? What about physical disabilities (though I recognize that these may have developed after college)? These answers, sadly, are likely to be relegated to the appendix.

I was intrigued that a positive coefficient appeared sometimes for same-sex partnered women. I understand the desire to not make too much of a result that only appears sometimes in the ACS and not in the NHIS. However, I would at least like to know which controls lead to this coefficient appearing. Particularly as there seems like there is a (very week) negative relationship between the fraction of women in same-sex couples in a field and the fraction of women in general.

Finally, pure speculation, looking at the chart of fraction of women in a field and the fraction of women in same-sex couples in a field, I wondered about the “academicness” of fields. Something like “Nuclear, Industrial Radiology, and Biological Technologies” is not only a small field, but very vocational. It prepares people to do power plant and infrastructure work, it is not a field that appears at any LACs or that there exist PhD programs in. This may be an extreme outlier but looking at the appendix graphs of women in same-sex couple I wonder about some pattern, as the fields with more women in same-sex couples seem to be in more “academic” fields. I partly wonder about this because of the way gender roles are expressed in the “occupational hierarchy” in that occupations that require less education tend to be more gender-segregated. I don’t know if this holds for people who are choosing more vocational majors, but I suspect it does. I wonder if less vocational fields are more or less likely to be chosen by those in a same sex couple.

Reviewer #3: Well-done analysis and clearly written. It would be good to have a brief discussion of the definition of STEM. The details can stay in the Appendix, but the paper needs a little bit on this. Also, do a check for typos.

6. PLOS authors have the option to publish the peer review history of their article (what does this mean?). If published, this will include your full peer review and any attached files.

Reviewer #1: No

Reviewer #2: **Yes: **Elisabeth Ruth Perlman

Reviewer #3: No

---

## [Author Response · Author response to Decision Letter 0]

15 Oct 2020

We have uploaded a separate file with detailed responses to reviewers.

---

## [Editor Report · Decision Letter 1]

19 Oct 2020

Turing’s children: Representation of sexual minorities in STEM

PONE-D-20-24017R1

Dear Dr. Carpenter,

We’re pleased to inform you that your manuscript has been judged scientifically suitable for publication and will be formally accepted for publication once it meets all outstanding technical requirements.

Kind regards,

Joshua L Rosenbloom

Academic Editor

PLOS ONE
---

## [Editor Report · Acceptance letter]

27 Oct 2020

PONE-D-20-24017R1 

Turing’s children: Representation of sexual minorities in STEM 

Dear Dr. Carpenter:

I'm pleased to inform you that your manuscript has been deemed suitable for publication in PLOS ONE. Congratulations! Your manuscript is now with our production department. 

Kind regards, 

on behalf of

Dr. Joshua L Rosenbloom 

Academic Editor

PLOS ONE